# Pattern mining and prediction techniques for user behavioral trajectories in e-commerce

**Xin Wang**[1]*, **Dong-Feng Liu**[2]

**1** School of Digital business, Nanjing Vocational College of Information Technology, Nanjing, China,
**2** Jiangsu Vocational College of Electronics and Information, Huai'an, China

* 53710198@qq.com

## Abstract

The trajectory of a user's continuous online access, which manifests as a sequence of dynamic behaviours during online purchases, constitutes fundamental behavioural data. However, a comprehensive computational method for measuring trajectory similarity and thoroughly analyzing user behaviour remains elusive. Analyzing user behaviour sequences requires balancing detail with data reduction while addressing challenges such as excessive spatial complexity and potential null results in predictions. This study addresses two critical aspects: First, it evaluates similarity in the time dimension of user behaviour sequence clustering. Second, it introduces a frequent sub-trajectory mining algorithm that emphasizes the order of user visits for trajectory analysis and prediction. We employ a variable-order Markov model to manage the growth of probability matrix size in forecasts. Additionally, we improve prediction accuracy by aggregating the time spent on specific web pages.

## 1. Introduction

Fueled by the Internet, advanced big data, and cloud computing technologies, we have entered a rapid proliferation of information. The extensive utilization of these technologies significantly enhances information resources and elevates the societal significance of information in people's everyday lives. Over time, from the inception of e-mail to the rapid growth of social media in the present day, the behavioral patterns of Internet users have undergone a significant transformation. The Internet has ubiquitously penetrated all facets of our professional, commercial, and even everyday existence. Nevertheless, this exponential expansion of information also presented novel obstacles. While users derive considerable convenience from the Internet, they also have the challenge of information overload. Determining one's true interests in the vast amount of information has grown progressively challenging, a phenomenon called "selection phobia." Informative providers now face the pressing challenge of differentiating themselves in the intense competition and effectively capturing and retaining consumers' attention. The advent of prediction systems offers

**Data availability statement:** The data relevant to this study can be accessed from the zenodo repository at https://doi.org/10.5281/zenodo.15064002. The key code for this study is available on GitHub at https://github.com/wx88dfl/Pattern-Mining-and-Prediction-Methods-for-User-Behavior-Trajectories-in-E-Commerce.

**Funding:** This work was supported by the High-Level Talents Scientific Research Startup Project of Nanjing Vocational College of Information Technology (Grant No. YB20220602).

**Competing interests:** The authors have declared that no competing interests exist.

a novel approach to address this issue. Analyzing multi-dimensional data, including users' behavior, knowledge, and interests, provides personalized content prediction that can achieve mutually beneficial outcomes for users and information producers. The prediction system operates in conjunction with the main website and achieves its purpose by thoroughly analyzing user data.

The implementation of prediction algorithms in e-commerce platforms like Taobao, Jingdong, Dangdang, etc., has shown impressive outcomes, greatly enhancing the company's operational profitability. It has garnered considerable interest from both the industry and academics. Prediction algorithms, as the fundamental components of prediction systems, have consistently been a focal point of experimental investigation. While conventional prediction algorithms relying on data statistics may meet customers' requirements to some degree, they seem inadequate in addressing the growingly individualized needs of consumers in the present day.

The emergence of community e-commerce presents novel prospects and obstacles for the advancement of prediction systems. Community e-commerce differs from conventional e-commerce platforms by prioritizing the individually tailored requirements of customers. Its user groups are more stable and exhibit more frequent purchasing patterns. This necessitates that the prediction system can more precisely capture the individualized attributes of customers and offer more refined prediction services. The limited data nature of community e-commerce renders typical prediction algorithms based on big data ineffective. Therefore, it is necessary to investigate novel algorithms and approaches to address the tailored prediction requirements of community e-commerce.

This study aims to investigate the present state and difficulties of a personalized prediction system for community e-commerce, examine the distinctions between it and a conventional e-commerce platform prediction system, and provide potential remedies. The prediction system will assume a more prominent position in community e-commerce, yielding an enhanced consumer purchasing experience and concurrently generating increased commercial value for enterprises.

Analyzing personalized consumer interest is essential in the e-commerce industry to enhance marketing effectiveness and user pleasure. In their study, Bhatnagar et al. [1] investigated the phenomenon of user interest in how users use e-commerce websites and established the temporal scope of interest. Results indicate that personalized marketing is most effective when implemented during the time window of interest, particularly for users who access an e-commerce platform via a search engine. This time window is considerably longer than users accessing the platform through an online banner advertisement. Nevertheless, these studies often regard user interests as fixed, disregarding the dynamic and real-time characteristics of user interests. To capture real-time user interest, Ding et al. [2] presented a correlation between users' inherent usefulness and the probability of purchasing by analyzing the hierarchy of web pages. Based on the assumption that there is a positive correlation between the level of user interest and the likelihood of making a purchase, a hierarchical Bayesian model was employed to determine the real-time interest of individual users. These users were classified into two states: high intent and low intent.

Furthermore, Li and Ding [3] developed a linear utility function model to determine that an individual user's real-time interest is influenced not only by previous browsing behavior but also by the combined impact of web pages and marketing stimuli. Alternatively, Tam and Ho [4] contended that the contact between an individual consumer and an e-commerce platform can serve as a persuasive force to alter the consumers' implicit interests and impact decision-making based on the business objectives of the online retailer. A novel web page personalization approach is proposed, incorporating preference matching levels, page suggestion set sizes, and sorting cues. These elements enable e-commerce merchants to manipulate customization to cater to individual consumers' unique cognitive requirements.

The mining and analysis of customer purchase records in enterprise information systems is a crucial method for uncovering the underlying knowledge of consumer behavior. Utilizing this data, researchers can develop models to forecast consumer behavior and offer decision-making assistance to businesses. Liu et al. [5] employed diverse machine learning models to assess over 100 features from new consumer data collected during Taobao's 'Double Eleven' period. Ultimately, they achieved rather accurate prediction findings. The present study highlights the significance of forecasting the consumer count for extensive e-commerce events. This paper illustrates the application of machine learning concepts in comprehending and predicting consumer behavior throughout extensive e-commerce events. The study by Li and Zhang et al. [6] examined the repeat purchase intention of new customers. It showed that integrating the SMOTE algorithm with the Random Forest algorithm resulted in superior prediction performance compared to a single method. The present study emphasizes algorithm fusion's capacity to enhance predictions' accuracy.

In contrast, Shen et al. [7] employed a highly explanatory tree model derived from the Alibaba e-commerce platform to forecast the characteristics of recurring customers. The researchers developed the prediction model and provided a comprehensive explanation of the model, including feature importance, partial dependency graph, and decision rule parameters. The significance of this study is in its ability to create both predictive outcomes and a comprehensive comprehension of the results, playing a critical role in formulating corporate strategy.

The advent of sequential recommendation algorithms within the realm of technology has presented novel prospects for developing personalized recommender systems. Unlike conventional recommendation algorithms, sequence recommendation algorithms consider user behavior's temporal aspect and interaction's significance. They aim to uncover the user's implicit interaction intentions from sequenced past behavior, so accessing the user's dynamics and potential interest within a specific timeframe. Sequence recommendation algorithms research is categorized mostly into statistical machine learning and deep learning [8]. Classic sequence recommendation systems, such as Markov chain-based recommendation, use the user's history sequence to forecast the next item they would click on. Rendle et al. [9] integrated first-order Markov chains with matrix factorization (MF) to represent users' first-order sequence characteristics. He et al. [10] addressed the limitations of the first two models by integrating similarity computation with higher-order Markov chains.

Conversely, deep learning-based sequence recommendation approaches begin with a multilayer perceptron (MLP). They progressively incorporate neural network models that address issues in other fields and deploy them to recommendation algorithms [11]. Convolutional neural networks (CNNs), recurrent neural networks (RNNs), graph neural networks (GNNs), and attention mechanisms are examples of representative models. To illustrate, the literature [12] on Text-CNN [13] improves the connections between features of items by combining the convolutional outcomes in various orientations. The GRU4Rec [14] and GRU4Rec+ [15] models are built upon Gated Recurrent Units (GRUs) [16], which identify a user's probable intention by analyzing their interaction sequences and making predictions about their future output.

Furthermore, Chang et al. [17] employed metric learning to reconstruct the user's feature representation. The SASRec [18] algorithm is founded on the self-attention mechanism, effectively utilizing the user's interest weight in the present interaction. The Ali team's BERT4Rec [19] is built upon the bidirectional Transformer [20], showcasing a robust processing and computation capacity for handling vast data. Similarity-based prediction methods improve prediction efficiency and reduce system processing by dividing users into separate groups and developing a trajectory prediction model for each group [21,22] These approaches' primary focus is to determine the distance between trajectories efficiently. Examples of

such methods include Dynamic Time Warping (DTW) [23], Longest Common Subsequence (LCSS) [24], Edit Distance [25], and Euclidean Distance [26]. Agrawal et al. [27] performed comprehensive experimental investigations on current trajectory distance algorithms. Meanwhile, Liu et al. [28] employ social propagation theory to categorize users based on the similarity of their mobile user trajectories and develop a second-order Markov trajectory prediction model. Quehl and colleagues [29] introduce a trajectory prediction approach that integrates trajectory similarity and heuristics to determine automatic decision-making parameters. This method is distinguished by its minimal spatio-temporal complexity.

In the study domain of user behavior trajectory prediction, existing methods depend on data-driven analytics to make predictions by analyzing user patterns in trajectory sequences. Nevertheless, these approaches have several evident limitations:

1. **Delimitations of User Behavior Analysis:** Current approaches frequently lack comprehensiveness and scope in evaluating user behavior, limiting the accurate comprehension of user behavior. Simultaneously, they disregard the significance of semantic and temporal characteristics, which are essential for enhancing the accuracy of predictions.

2. **User clustering complexity:** The computation technique of similarity in user clustering is highly intricate. Current approaches typically necessitate the human specification of feature dimensions for various websites, which not only lacks universality but also presents challenges in terms of automation.

3. **Prediction accuracy limitations:** Current approaches often fail to include users' past trajectories in the user behavioral context while predicting new trajectories, significantly decreasing prediction accuracy.

4. **Challenges in computational efficiency and results:** Several current approaches for forecasting user behavioral trajectories encounter issues related to significant computational resources and incorrect prediction outcomes, which greatly impact the overall effectiveness of these systems.

In response to these difficulties, this paper presents a set of novel approaches designed to enhance the precision and effectiveness of predicting user behavior trajectories:

1. **In-depth multi-dimensional feature analysis:** this work explores the geographical, temporal, and semantic characteristics of user behavioral trajectory data and presents a formal model for representing user trajectory sequences.

2. **Advanced clustering algorithm:** This paper presents an innovative clustering technique that utilizes clickstreams and custom events to enhance comprehension and analysis of user clicking activity. The overall dataset is first subjected to density-based clustering in the clustering procedure. Subsequently, the clustering results of the resulting multiple periods are combined based on the origin of the points in the clustering results. This integration ensures the consistency of the coding rules, enabling direct comparison and computation of the clusters. Furthermore, by partitioning user sessions into distinct session clusters, we may effectively discern user behavior patterns more precisely.

3. **Trajectory prediction method based on Hidden Markov Model:** We present a Hidden Markov Model-based trajectory prediction approach to address the limitations of current methods in handling semantic information in trajectory data. This model combines spatial and semantic variables to increase the accuracy of trajectory prediction.

   1) We exploited the similarity clustering technique of trajectory points and the transformation algorithm of common sub-trajectories to extract the ideal hidden state of the model effectively.

   2) Utilizing semantic properties, we made predictions about the regions visited by users by analyzing the semantic characteristics of mobile trajectories. Additionally, we verified the relationship between users' clicking behavior on web pages and their purchase intention.

   3) By integrating trajectory point and area types prediction models, we enhance the precision of predicting the user's next destination, greatly improving the forecast's accuracy.

The paper is organized as follows. Section 2 is devoted to foundational principles. In Section 3, we evaluate clustering techniques for page-type sequences. Section 4 introduces a time-space semantics-based approach for trajectory prediction. Section 5 illustrates experimental analysis. The paper ends with the conclusions and future work in Section 6.

## Methods

This study involves the behavioral analysis of users and the application of data mining techniques.

1. The data used in this study is derived from publicly available datasets, ensuring that all research activities are conducted within the scope of regulated consent.

2. Anonymized information are utilized to conduct the research, ensuring the privacy and confidentiality of the data subjects.

3. This research does not pose any harm to human participants and does not involve sensitive personal information or commercial interests. The study adheres to the principles outlined in the Declaration of Helsinki. Furthermore, it complies with the ethical exemption requirements specified in the "Ethical Review Measures for Life Science and Medical Research Involving Humans" promulgated in China, qualifying it for an exemption from ethical review.

## 2. Foundational principles

In the context of web pages, a clickstream record refers to a comprehensive series of sequentially arranged web pages that a person consumes during their visit to a website. A clickstream record is a comprehensive series of sequentially arranged application pages a user loads while accessing an application. It includes access details for each page, such as the page name, load time, and dwell time. It also provides overall information such as the order in which the pages load, the number of pages, the access time, and the total time.

### 2.1 User behavior trajectories.

Given the clickstream data acquired without altering the application source code, the user session record based on clickstream can be abstracted in the following manner.
　　Abstracted user session record:
　　**Definition 1:** A series of user behavior trajectories is conceptualized as follows:

$$P^{T_j} = \left\langle \left( p_1^{T_j}, t_1^{T_j} \right), \left( p_2^{T_j}, t_2^{T_j} \right), \ldots, \left( p_i^{T_j}, t_i^{T_j} \right), \ldots, \left( p_n^{T_j}, t_2^{T_j} \right) \right\rangle. \tag{1}$$

Let's $P^{T_j}$ indicate the sequence of user behavioral trajectories in the period $T_j$, where the ith page $p_i^{T_j}$ viewed by the user during that period is followed by the duration of time $T_j$ spent on that page, and generally satisfies $T_j = \sum_{i=1}^{n} t_i^{T_j}$: Null describes an empty page $p_n^{T_j}$, indicating the termination of the user's access to that page.
　　**Deduction:** Dynamic pattern recognition of user behavior trajectory sequence is primarily achieved by selecting suitable statistical indicators and determining the accurate time observation granularity. The changes in the user's behavioral patterns are then described and calculated:

$$PT_D = \left\{ P^{T_1}, P^{T_2}, \ldots, P^{T_j}, \ldots, P^{T_N} \right\}. \tag{2}$$

The formula provided above denotes a compilation of sequences characterizing user behavior trajectories.
　　The definition provided above refers to the page as a loaded object for analysis. We classify page types in the context of online shopping, as defined in Table 1:

**Table 1. Types of web pages for shopping and abbreviations.**

| Abbreviation | Web Page Class Name | Definition of categorization |
|---|---|---|
| H | Home | Visit the home page |
| A | online account | User login, change address, view order status, etc. |
| S* | Payments* | payment page |
| D | Add to cart | Add to Cart / Favorites |
| F | Favorites/Cart | View Shopping Cart / View Favorites |
| G* | Products * | Clicked a product in the product list |
| R | Evaluation | View Product Reviews |
| B | Brand | Official Flagship Store/Click Brand |
| P | prices | Click on the price / follow the price range |
| Y | character | Click on popularity |
| V | sales volume | Click sales |
| T | Product Properties | Product Specific Information |
| C | catalogs | Search Result / Product List |
| O | else | Other pages |

The typing * indicates that pages of specific product categories are nested under the type page. Therefore, the type sequence of the corresponding page is defined as follows:

$$L^{T_j} = \left\langle \left( l_1^{T_j}, t_1^{T_j} \right), \left( l_2^{T_j}, t_2^{T_j} \right), \dots, \left( l_i^{T_j}, t_i^{T_j} \right), \dots, \left( l_n^{T_j}, t_n^{T_j} \right) \right\rangle,$$

(3)

where $L^{T_j}$ denotes the period $T_j$ in which of the page type behavioral trajectory sequence, $l_i^{T_j}$ represents the ith page type visited by the user in the period the ith page type visited by the user in the period, the time spent on the visited page is and satisfies $\sum_{i=1}^{n} t_i^{T_j}$. The time spent on the page is the time that satisfies. Using the type sequence representation of a page can narrow down the number of page-type sequences and simultaneously reduce the complexity of the subsequent analysis.

Preprocessing of the type sequence of the page

User behavior data needs to be preprocessed. The processing rules for the type sequence of the page are shown below:

1) If the time of the type sequence of the pages $t_1^{T_j}, t_2^{T_j}, \dots, t_i^{T_j}, \dots, t_n^{T_j}$ is less than $t_1'$, then it means that the user may have clicked on the page by mistake or is not interested in the page, and therefore, the page should be removed from the type sequence. In the experiment, the reasonable threshold $t_1'$ can be set according to the actual need, such as 15 seconds;

2) If the time of the type sequence of the page $t_1^{T_j}, t_2^{T_j}, \dots, t_i^{T_j}, \dots, t_n^{T_j}$ is greater than $t_2'$, then it means that the page probably failed to load or the user has left, so the page should be removed from the type sequence of the page. In an experiment, the reasonable threshold $t_2'$ can be set according to the actual need, for example, 30 minutes;

3) If the type sequence of pages is longer than (page repeatable) $t_3'$, then it means that the page sequence was generated aimlessly by the user. Therefore, the sequence has no value and needs to be removed from the data. In the experiment, N is set to 30. When there are duplicate sub-page sequences, they can be represented as super nodes to

simplify the type sequence of the page. As shown in the figure below, the dotted line in the middle of the figure is the common duplicate page type sequence $A \rightarrow S\_A1 \rightarrow D$, writing this sequence of three-page nodes as super node W. An example of super node merging is shown in <u>Fig 1</u>, where the nodes within the dashed line are merged into super node w:

A null end page is included in the user session to determine the residence time of the last page. The load time of that page is then considered as the end time of the series. The page's type sequence can be split, with the option to exclude irrelevant page types and extract the pertinent page types and dwell periods from the existing page's type sequence into a new page's type sequence based on the product type of the page where the order was made.

## 2.2 Similarity between type sequences

Without concern for the dwell time, our focus is solely on the order of consecutive visits to the visited pages. Compounding the similarity between two distinct access sequences is synonymous with calculating the similarity between two strings, which can be accomplished by employing the largest common substring (LCS). The largest common substring (LCS) is the largest among all common substrings. It is possible to classify substrings into two distinct groups: continuous and discontinuous. Discontinuous substrings can be derived by extracting multiple items from the initial sequence. Hence, the conventional dynamic programming method is applicable for solving the non-contiguous maximal common substring. Consider two-page type sequences represented as follows:

$$L^{T_j} = \left\langle \left(l_1^{T_j}, t_1^{T_j}\right), \left(l_2^{T_j}, t_2^{T_j}\right), \ldots, \left(l_i^{T_j}, t_i^{T_j}\right), \ldots, \left(l_n^{T_j}, t_n^{T_j}\right) \right\rangle,$$

(4)

$$L^{T_k} = \left\langle \left(l_1^{T_k}, t_1^{T_k}\right), \left(l_2^{T_k}, t_2^{T_k}\right), \ldots, \left(l_i^{T_k}, t_i^{T_k}\right), \ldots, \left(l_n^{T_k}, t_n^{T_k}\right) \right\rangle.$$

(5)

Split the above page type sequence into two parts, i.e., page type sequence and dwell time sequence:

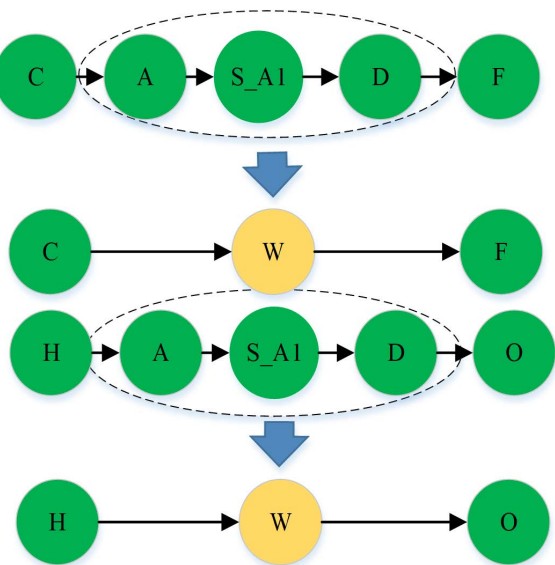

**Fig 1. Exemplification of node consolidation on a webpage.**

$$L^{T_j} = \left\langle l_1^{T_j}, l_2^{T_j}, \ldots, l_i^{T_j}, \ldots, l_n^{T_j} \right\rangle,$$ (6)

$$T^{T_j} = \left\langle t_1^{T_j}, t_2^{T_j}, \ldots, t_i^{T_j}, \ldots, t_n^{T_j} \right\rangle,$$ (7)

$$L^{T_k} = \left\langle l_1^{T_k}, l_2^{T_k}, \ldots, l_i^{T_k}, \ldots, l_n^{T_k} \right\rangle,$$ (8)

$$T^{T_k} = \left\langle t_1^{T_k}, t_2^{T_{jk}}, \ldots, t_i^{T_k}, \ldots, t_n^{T_k} \right\rangle.$$ (9)

Sequence derived from two different page type sequences:

$$L^{T_j} = \left\langle l_1^{T_j}, l_2^{T_j}, \ldots, l_i^{T_j}, \ldots, l_n^{T_j} \right\rangle,$$ (10)

$$L^{T_k} = \left\langle l_1^{T_k}, l_2^{T_k}, \ldots, l_i^{T_k}, \ldots, l_n^{T_k} \right\rangle.$$ (11)

The public page type sequence is extracted in:

$$U = \langle u_1, u_2, \ldots, u_i, \ldots, u_N \rangle$$ (12)

The corresponding time dwells are expressed as follows:

$$T_1 = \left\langle t_1^1, t_2^1, \ldots, t_i^1, \ldots, t_N^1 \right\rangle,$$ (13)

$$T_2 = \left\langle t_1^2, t_2^2, \ldots, t_i^2, \ldots, t_N^2 \right\rangle.$$ (14)

Introduce the notion of access depth, which involves determining the access depth for each page type in the sequence of the initial visit to the serial number. It applies if the sequence of future visits remains the same as the first visit to the main, for instance, sequence $A \to B \to C \to C \to D$. A's access depth is 1, B's is 2, C's is 3, and C's access depth remains three on the second instance. The following graph displays the series of access depths:

$$F_1 = \left\langle f_1^1, f_2^1, \ldots, f_i^1, \ldots, f_N^1 \right\rangle,$$ (15)

$$F_2 = \left\langle f_1^2, f_2^2, \ldots, f_i^2, \ldots, f_N^2 \right\rangle.$$ (16)

The temporal similarity of each visit to the public page is as follows:

$$ST_i = \frac{\min\left(t_i^1, t_i^2\right)}{\max\left(t_i^1, t_i^2\right)}.$$ (17)

The similarity in depth for each visit to the public page is as follows:

$$FT_i = \frac{\min\left(f_i^1, f_i^2\right)}{\max\left(f_i^1, f_i^2\right)}.$$ (18)

Apparently, there are $0 < ST_i, FT_i \leq 1$. If any two users stay on the same page for a similar amount of time as well as depth, it means that the closer the similarity of their interest in that page is (i.e., the closer it is to 1). Therefore, the similarity calculation of two different users A and B on the same visited page type can be defined as follows

$$ST_c = \frac{1}{N} \sum_{i=1}^{N} ST_i, \tag{19}$$

$$FT_C = \frac{1}{N} \sum_{i=1}^{N} FT_i. \tag{20}$$

Finally, the respective similarity is calculated as:

$$SI_1 = ST_c \times \left( \frac{\sum_{i=1}^{N} t_i^1}{T_j} \times \frac{\sum_{i=1}^{N} t_i^2}{T_k} \right)^2, \tag{21}$$

$$SI_2 = FT_c \times \left( \frac{\sum_{i=1}^{N} f_i^1}{F_j} \times \frac{\sum_{i=1}^{N} f_i^2}{F_k} \right)^2. \tag{22}$$

where $F_j$ and $F_k$ is the sum of the depths of the original page type sequences. The total similarity is defined as follows:

$$SI = SI_1 \times SI_2. \tag{23}$$

## 3. Clustering techniques for sequences of page types

To streamline the calculation of cluster coding, which is both computationally demanding and time-consuming, this section explores the appropriate clustering schemes to reduce the complexity of sequence analysis further. The datasets from various periods can be merged into a unified dataset. Subsequently, density-based clustering can be applied to the entire dataset. The clustering results can be further categorized depending on the origin of the calculation points and, after that, segregated. Furthermore, suppose the number of points in the clustering clusters derived from separation is lower than the minimum number of points in the cluster of cluster analysis. In that case, the cluster label must be substituted with Noise. Conversely, it is possible to group and partition the sequences into several independent clusters to simplify the examination of sequences. Nevertheless, the conventional DBSCAN (Density-Based Spatial Clustering of Applications with Noise) method primarily increases the size of the clusters by considering the density of the surrounding environments, which refers to the amount of other objects in the vicinity.

There exist two primary issues with the algorithm:

This paper introduces the enhanced DBSCAN algorithm, which has been modified to address two specific challenges: (1) Sequence similarity cannot be quantified using the conventional distance similarity function and requires modification; (2) Sequences with multiple sub-sequences may exhibit similarity, leading to challenges in determining classification clustering sub-clusters;

(1) Substitute ε-neighborhood with r-neighborhood, where r ($0 \leq r < 1$) represents the threshold for sequence similarity, and r-neighborhood indicates an object space where all objects in that space are greater than r similar to the core object O.

The computation may be performed using the formula above.

(2) The type sequence of the page is pre-processed by splitting the category sequence. It ensures that the sequence is placed in the cluster with the highest similarity, enhancing the cluster formation conditions. The cluster's number of elements determines the initial criteria for cluster formation. If the number of elements in the cluster is too small, it will be

classified as NOISE. The present study focuses on the sequence unit and aims to provide a higher level of processing granularity. Consequently, the criteria for cluster formation are adjusted. Specifically, if the number of common elements in the r-neighborhood of two clusters is less than λ times of the smaller cluster, merging the two clusters is not possible. The number of λ times of the smaller cluster is defined as $Nmin_m$.

---

Clustering algorithm for different time periods based on cluster analysis
Input: $TT_1$, ...., $TT_j$, ...., $TT_N$ Time period dataset $PT_1$, ...., $PT_j$,...., $PT_M$ , similarity threshold r, merging threshold λ
1: Consolidation $PT_1$, ...., $PT_j$, ...., $PT_M$ Get the overall dataset PTS
2: Perform clustering calculation RDBSCAN (PTS, r, λ)
3: for each cluster of the overall dataset PTS i do
4: for each sequence $L_j$ in cluster i do
5: if $L_j \in PT_1$, ...., $PT_j$ , ...., $PT_M$ then
6: respectively $L_j$ Labelled with clusters j
7: for Dataset $PT_1$ , ...., $PT_j$ , ...., $PT_M$ for each cluster m do
8: if the number of points in cluster m is less than $Nmin$ then
9: Label cluster m as Noise
Returns: clustering results for datasets

---

The clustering algorithm is as follows:

---

RDBSCAN (PTS, r, λ) algorithm
Inputs: overall dataset PTS, similarity threshold r, merge threshold λ
1: Mark all objects in the PTS as not accessed
2: Define S as a cluster set and initialize S as empty
3: For each unvisited object in the PTS p
4: Mark p as visited
5: If the r-neighborhood of p contains at least $Nmin$ objects
6: Create a new cluster $Cl_m$ and add the objects in p and p's r-neighbourhood to the $Cl_m$
7: Add $Cl_m$ Add S
8: END FOR
9: FOR Every set in S $Cl_m$ and $Cl_n$ (m≠n)
10: If the common elements of $Cl_m$ and $Cl_n$ are more than λ times of $Cl_m$ or $Cl_n$
11: then merge $Cl_m$ and $Cl_n$ into a new cluster and call it C. Also add C to S.
12: ELSE $Cl_m$ and $Cl_n$ into two separate clusters
13: END FOR
14: Output S

---

After executing the above algorithm, the clusters for clustering are named, and after naming the clusters, the clusters are merged when the end time of the sequence in the cluster is less than **SE** from the start time. Serialize each trajectory and formalize it into the following structure:

$$A_1 \xrightarrow{D_1} A_2 \xrightarrow{D_2} \ldots \to A_i \xrightarrow{D_i} \ldots \xrightarrow{D_{N-1}} A_N, \tag{24}$$

where $A_1, A_2, \ldots, A_i, \ldots, A_N$ is the renaming of the cluster in which the sequence is located, referred to as a trajectory node in the trajectory sequence, and $D_1, D_2, \ldots, D_i, \ldots, D_{N-1}$ is the period of the interval between neighboring nodes of the sequence, i.e., the difference between the end time of the neighboring node and the next start time.

## 4. A time-space semantics-based approach for trajectory prediction

### 4.1 Frequent trajectories generation.

This approach is founded on trajectory sequences, which necessitate the integration of the trajectory points within the sequences, together with the semantic and temporal characteristics of the scene during the merging process. Firstly, the terminology of support in a trajectory sequence is introduced.

**Definition** 2 Support The support of a trajectory node in a dataset with n trajectory data is equivalent to the count of trajectories to which the trajectory node is associated.

Frequent trajectory tree generation is defined as follows

To build a frequent trajectory tree, it is necessary to create tree-like storage by systematically traversing the past trajectory data.

The nodes' attributes can be adjusted to meet the user's specifications, and the time intervals between adjacent nodes are indicated on the tree structure of the nodes. The tree structure is constructed by traversing all trajectory data sets once. The specific structure construction process is as follows:

1) From the beginning of a trajectory, nodes in the trajectory are turned into branches of the tree, and nodes are treated as bifurcation nodes in the tree when the node of the trajectory serves as a common node for multiple trajectories;

2) In the process of constructing the tree, the corresponding node name, support, and pointer of the node are constantly placed. When the node in the trajectory is constructed in the tree for the first time, the pointer is pointed to that node, and the same node appearing later is pointed to the next occurrence of that node in the tree, in the order of construction;

3) The interval times in the trajectory are labeled on the branches in the tree, and the times are labeled in the sequences.

An illustrative sequence of three trajectories is shown as follows:

$$A_1 \xrightarrow{13} A_2 \xrightarrow{4} A_3 \xrightarrow{8} A_4 \xrightarrow{7} A_5, \tag{25}$$

$$A_1 \xrightarrow{12} A_4 \xrightarrow{8} A_3 \xrightarrow{9} A_1, \tag{26}$$

$$A_3 \xrightarrow{13} A_2 \xrightarrow{4} A_5 \xrightarrow{5} A_4. \tag{27}$$

The tree trajectories are arranged in a series, and a table is established on the periphery of the tree structure trajectory. The table's first column represents the point's name, the second column represents the value of the support, and the third column marks the pointer. The example is shown in Fig 2. The sequence of three trajectories is merged and reorganized into a tree structure.

Frequent Pattern Mining: Frequent trajectory mining is performed by partitioning the trajectories into three subgroups: the candidate trajectory Can [], the frequent trajectory F [], and the final trajectory R []. Firstly, all the trajectory points that have support over the minimal support level should be identified. Next, all the sub-trajectories that meet the specified criteria as the endpoints are identified and combined into Can []. Step three: Verify all the combined sub-trajectories in Can [] to determine if they exceed the minimal support criterion and then transfer them to F []. In the fourth stage, verify the sub-trajectories in F [] again, considering the user's time limitations to ensure that the requirements are satisfied for inclusion in R[].

The approach described in this section utilizes an array to hold the conditional tree. During the traversal of the tree structure, the sub-trajectories obtained are stored in the array, enhancing the mining efficiency. The conventional algorithm for mining frequent trajectory patterns only computes the highest frequency sub-trajectories. However, this algorithm cannot satisfy the need to increase the frequent sub-trajectories during dynamic trajectory prediction. Therefore, this method is enhanced to generate frequent sub-trajectories within a specified frequency range tailored to various requirements.

## 4.2 Calculation of state transfer matrix.

Through the examination and analysis of those above, using the frequent sub-trajectories and combining the trajectory points derived from these sub-trajectories, it becomes feasible to calculate the likelihood of reaching the desired

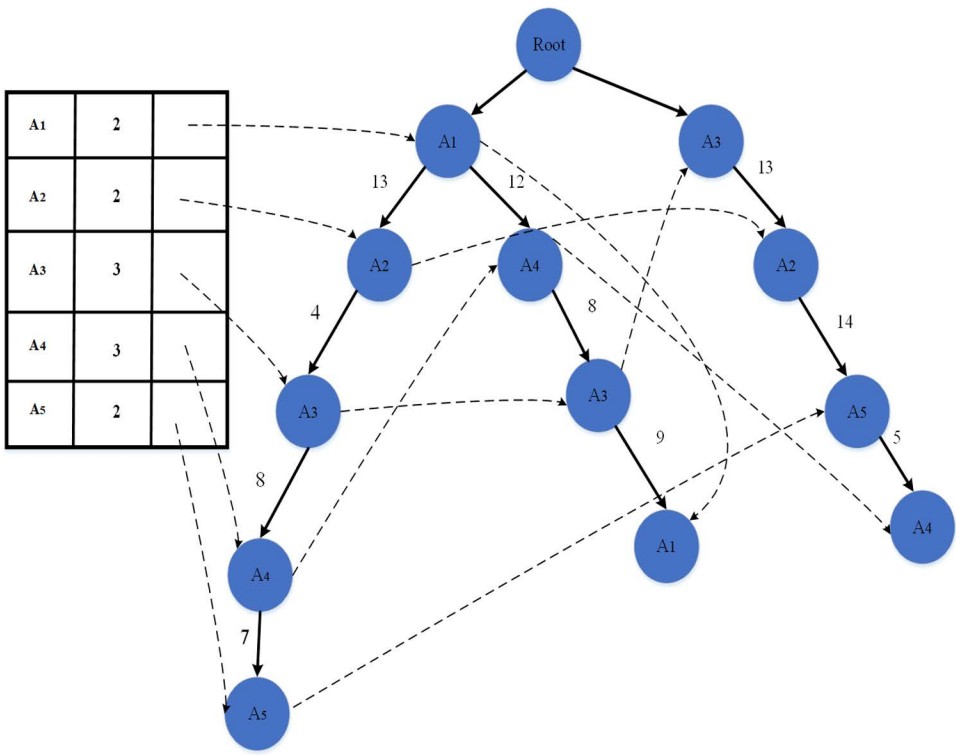

**Fig 2.** Sequence of tree trajectories.

destination that the user is on the verge of achieving based on the state transfer matrix within the trajectory. This section exclusively addresses the computation of the state transfer matrix.

A Markov process of higher order: This work presents a higher-order Markov process to address the variability in the probability of a trajectory reaching the current node, which factors beyond the current node position may influence. In contrast to the first-order Markov process, the higher-order Markov process considers the correlation between the positions of the k preceding historical nodes visited by the user and the target node to be visited. Despite its ability to more accurately replicate the user's visit trajectory, there are two challenges in considering the trajectory of the historical visited nodes. Firstly, the size of the transfer matrix increases exponentially. Secondly, the training data set may not include historical data, resulting in null results for the visit. Thus, in this study, we employ a Markov process with variable order to forecast the outcomes by analyzing the paths of various historical dates. Because considering the last two positions results in more transfer combinations. For instance, when using second-order Markov modeling, it is necessary to construct a larger transfer probability matrix. Therefore, the probability of moving the user's next access position when predicting the user's access trajectory via second-order Markov modeling can be specified as:

$$(A_i | A_{i-1}, A_{i-2}) = \frac{N(A_{i-2}, A_{i-1} \to A_i)}{\sum_{k=1}^{N} N(A_{i-2}, A_{i-1} \to A_k)},$$

(28)

where the numerator represents the distance in the training dataset from node $A_{i-2}$ through node $A_{i-1}$ to node $A_i$ in the training data set, and the denominator represents the number of all possible passages from node $A_{i-2}$ as the starting point passing through node $A_{i-1}$ to the number of nodes to each node.

Given the potential issue of null prediction outcomes in the conventional Markov model, we propose using a variable-order Markov model with partial matching rules. This work presents a method that uses a frequent tree structure to address the transfer probability matrix's sparsity issue. It minimizes the number of model orders by imposing an upper limit on the order value. Additionally, the method incorporates an escape mechanism to handle the problem of null prediction probability. The escape method is employed for computation, whereby the user designates the sub-trajectory to be seen and observes the sub-trajectory. The next node in the trajectory is determined by differentiating between the sets $\sum \pi$ and $\sum \pi'$, which represent the sub-trajectory and the set of next nodes in the immediate vicinity in the sub-trajectory $\pi$, respectively. The sub-trajectory $\pi$ nodes are those not part of the set of immediately next nodes in the sub-trajectory. Determine the probability assignment of $P(E|\pi)$ and, consequently, the probability of $\sum \pi'$. The allocated probability is denoted as $1 - P(E|\pi)$. The following formula may express the probability:

$$P(e|\pi) = \begin{cases} P'(e|\pi) & e \in \sum \pi \\ P'(E|\pi) \cdot P'(e|\pi') & others \end{cases},$$

(29)

where $\pi'$ denotes the $\pi$ the suffix information after that. The other probabilities are defined as follows:

$$P'(e|\pi) = \frac{C(\pi e)}{C(\sum \pi) + \sum_{e \in \sum \pi} C(\pi e')},$$

(30)

$$P'(E|\pi) = \frac{C(\sum \pi)}{C(\sum \pi) + \sum_{e \in \sum \pi} C(\pi e')}.$$

(31)

During the model learning phase, the tree structure T is initially built using the frequent trajectory data from the training set. If the set model

Given a maximum order of N, the tree's height is N + 1. Analogous to the binary tree structure, the nodes are designated with labels indicating the frequency of their occurrence in the trajectory. The branches of the tree consist of sequences of trajectories, denoted as $P'(e|\pi)$ and $P'(E|\pi)$. The calculation is executed based on the precise location of each step in the recursive computation.

### 4.3 Forecasting the trajectory of shop categories.

Category prediction using Markov models: Given a user's visits to n periods $T_1, T_2, \ldots, T_n$, a, a prediction is generated about the type of shop in the next period using Markov types. Once a user accesses a web page, the subsequent web page may be influenced by pertinent tags and ribbon settings on the current page. The next visited web page is often accessed by clicking on the current page, which opens the following one.

Given a sequence of trajectories:

$$A_1 \xrightarrow{D_1} A_2 \xrightarrow{D_2} \ldots \rightarrow A_i \xrightarrow{D_i} \ldots \xrightarrow{D_{N-1}} A_N.$$

(32)

The number of possible shop type sequences that can be obtained is:

$$NP = \prod_{i=1}^{n} \left| category_{A_i} \right|.$$

(33)

For each point in the sequence $A_i$, which may contain multiple commodity types accessed by the user, multiplying the number of all commodity types indicates that the possible trajectories are traversed according to the commodity types. In contrast, the user traverses the sequence of trajectories.

**Shop Type Weighting Calculation**

Input: Sequence of candidate trajectories $X_1$ Predicted set of possible commodity types $H(i)$

Output: Weights of shop types for candidate trajectory sequences

1: $\beta = \frac{c(i)}{len(X_1)}$ , $W_{category}[i] = [\varnothing]$ //initialization

2. for $A_i$ in $X_1$ do //Candidate Trajectory Sequence

3. for $G(A_i)$ do

4. if $G(A_i)$ in $H(i)$ then // determine if the shop type covered by the candidate location is in the returned result

5. $W_{category}[i] += \beta$ // Add shop type weighting for wireless connection points

6. end

7. end

8. end

9. Return $W_{category}[i]$

Type of shop contained in the ith node is denoted by $G(A_i)$.

## 4.4 Candidate trajectory prediction algorithm

A spatial semantic-based prediction approach for Hidden Markov page trajectories is developed after selecting the hidden states, finding the ideal hidden state sequence solution, and predicting the type of shop that will be visited. Each node in the trajectory sequence computes the visit probability after merging the shop kinds. The parameter controls the item type ratio and the node size in the resulting output. The size ranges from 0 to 1, as determined by the following formula:

$$PN(A_i) = (1 - \gamma) \cdot GM(A_i) + W_{category}[i]. \tag{34}$$

The model first initializes the parameters on line 1 of the algorithm. It then determines the length of the trajectory sequence on lines 2–3 of the algorithm. Viterbi's algorithm decodes the optimal sequence of hidden states on line 5 for the given sequence of trajectories. Subsequently, all the nodes that may visit the next node are traversed, and the probability of their possible visit is calculated. Then, the nodes are ranked on lines 6–9 of the algorithm. Finally, the algorithm returns the candidate nodes.

## 5. Experimental analysis

An e-commerce application has been chosen as the test target. A Shenzhen-based information technology business is responsible for developing the App. The data-collecting function is designed within the App's source code, recompiled, and deployed on 28 test devices. Twenty-eight users are distributed to carry out focused testing on different business modules within the application. The level of detail in the information analysis of the clickstream data mostly focuses on the individual pages within the App and categorizes them based on their kinds. The pages users' access are gathered and transformed into page trajectories on a monthly cycle. These page trajectories are then further converted into page-type trajectories based on the page types. The total number of converted page-type trajectories is 1358. The neighborhood density threshold is defined as the natural logarithm of N, where N represents the number of trajectories. The similarity threshold is set to 0.7, and the merge threshold is 1/2.

The ratio inside a cluster is determined based on the count of page-type sequences included in the cluster.

The clustering clusters are initially encoded at the character level to streamline the mining and analysis of page-type sequences.

App can be effectively categorized into four modules based on their functions: shopping, communication, service, and personal center. During the experiment, several users performed simultaneous testing of several modules. The distribution of users' tests is depicted in Fig 3.

Additionally, the clusters have been renamed to streamline the sequence, as indicated in Table 2.

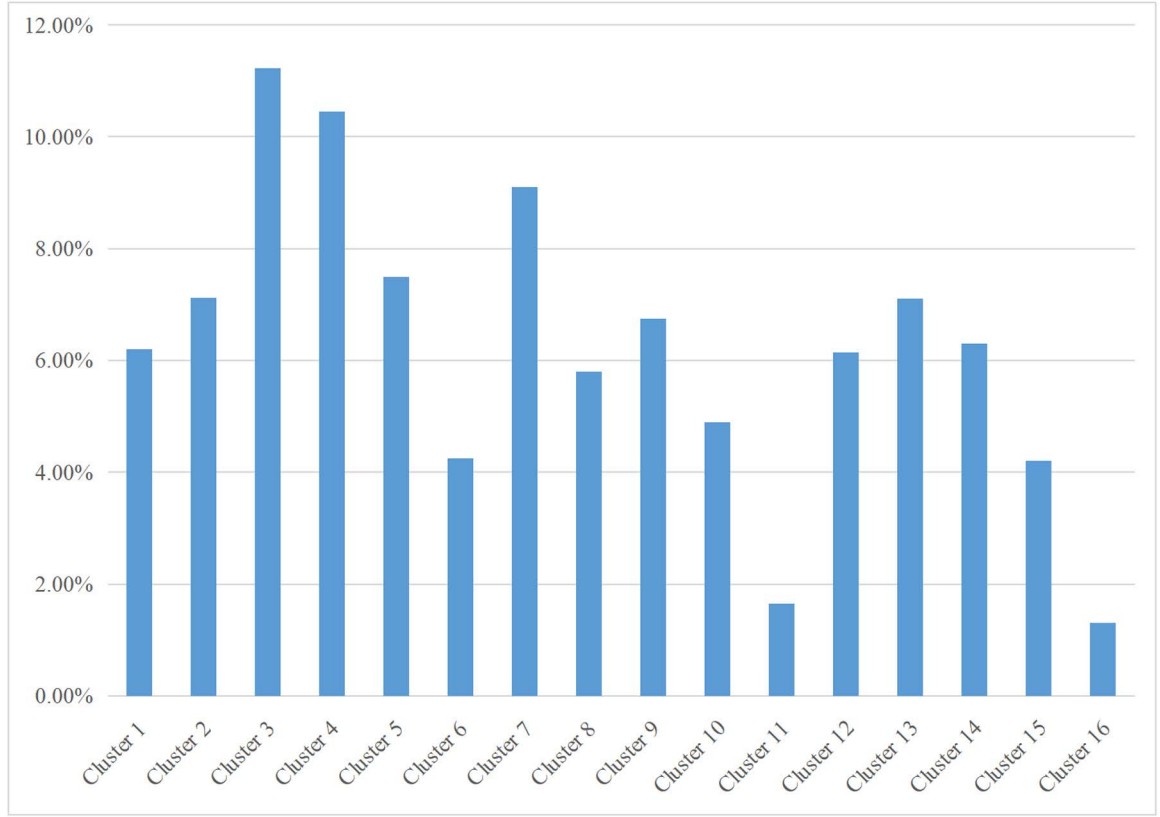

**Fig 3. Aggregate scale diagram of clusters.**

The 1358-page type trajectories were grouped into 16 clusters according to each user's trajectory. When the interval between two-page type trajectories was less than 1 hour, the trajectory sequence was reconstructed using clusters as nodes. The resulting trajectory sequence consisted of 348 trajectories.

The topic of this paper is evaluation indicators and comparison algorithms.

This work presents a prediction approach that calculates the likelihood of visiting each trajectory node. The forecast results are based on the top k candidate nodes with the highest probability. The trials employ the following evaluation measures [30] to assess the reliability of the prediction approach and validate its effectiveness:

(1) P&1 represents the likelihood that the candidate position with the highest probability is accurate.

(2) P&6 represents the likelihood that the accurate position is one of the first six candidate nodes generated by the prediction model.

This metric, known as Mean Reciprocal Rank (MRR), represents the average value of the reciprocal rank of the obtained results as follows:

$$MRR = \frac{1}{N} \sum_{i=1}^{N} \frac{1}{rank_i}.$$

(35)

The Mean Absolute Error (MAE) as an evaluation metric was not used in the tests because the trajectory prediction approach presented in this paper provides the labeled values of the trajectory nodes instead of the precise spatial coordinates.

Hence, this assessment measure is unsuitable for the research conducted in this publication.

**Table 2. Renaming of clusters.**

| Name of the cluster | Renaming of clusters |
| --- | --- |
| Cluster 1 | $A_1$ |
| Cluster 2 | $A_2$ |
| Cluster 3 | $A_3$ |
| Cluster 4 | $A_4$ |
| Cluster 5 | $A_5$ |
| Cluster 6 | $A_6$ |
| Cluster 7 | $A_7$ |
| Cluster 8 | $A_8$ |
| Cluster 9 | $A_9$ |
| Cluster 10 | $A_{10}$ |
| Cluster 11 | $A_{11}$ |
| Cluster 12 | $A_{12}$ |
| Cluster 13 | $A_{13}$ |
| Cluster 14 | $A_{14}$ |
| Cluster 15 | $A_{15}$ |
| Cluster 16 | $A_{16}$ |
| Noise | N |

The most accurate algorithms for trajectory prediction are the Markov Model Carousel (MMC), the Markov Model, and the Hidden Markov Model. The usefulness of the trajectory prediction method described in this paper is validated by experimental testing.

This chapter compares the trajectory prediction algorithm STS and five predefined benchmark methods.

The present study proceeds to conduct an empirical investigation of the suggested trajectory prediction model:

(1)  The Hidden Markov Model (HMM) is a well-referenced approach in trajectory prediction [31].

(2)  The LHMM technique is well-suited for the modeling and analysis of spatio-temporal trajectory data [32]

(3)  Mobile Machine Computing: capable of simulating the motions of mobile users [33].

(4)  Temporal Predictive Analysis (TPA) considers the time-related characteristics of the trajectory and can provide more precise forecasts of paths inside the campus [34].

(5)  STM: Trajectory similarity is considered [30].

(6)  The HST-LSTM is a trajectory prediction technique that utilizes long-short memory neural networks [35].

The present study proceeds to conduct an empirical investigation of the suggested trajectory prediction model:

1. We examine how various trajectories lengths impact the model's predictive performance.

2. We investigate the influence of the number and size of frequent sub-trajectories on the prediction performance.

3. We validate an experimental prediction model incorporating temporal, spatial, and semantic information.

The effects will be compared using the benchmark algorithm.

 

### 5.1 Influence of trajectory length on empirical findings.

Within the context of analyzing trajectory sequences of varying lengths using the Hidden Markov Model, this subsection primarily focuses on examining the impact of different trajectory lengths on the prediction accuracy of the Markov Model. 348 trajectory sequences were generated, with most trajectory lengths centered between 4 and 11. Therefore, we demonstrate in Table 3 the impact of trajectory length on the results.

The assessment metrics of the experiment remained rather consistent regardless of the variation in trajectory length. The P&1, the P&6 and the findings of MRR remained stable within the ranges of 13% and 14%, 36% and 41%, and 26% and 29%. The stability of the forecast performance of the Markov model for trajectory sequences of varying lengths is attributed to its independence from the complete historical trajectory. This characteristic aligns with the inherent nature of Markov models.

### 5.2 Impact of frequent sub-trajectories on the Accuracy of predictions.

Being common sub-trajectories that partially represent the preferences of users and encapsulate the group movement patterns, they significantly influence the click trajectories of users. Hence, this study primarily aims to validate the correlation between the magnitude and quantity of common sub-trajectories and the accuracy of model predictions. The investigations involve frequent sub-trajectories with sizes ranging from 3 to 6. The experimental results are depicted in Fig 4.

Fig 4 demonstrates that the model's predictive capability improves as the number of frequent sub-trajectories increases. It can be attributed to the reduction in states in the transfer probability matrix of the prediction method and the increase in matrix density, which mitigates the issue of null prediction results. Further evidence indicates that the model's predictive capability is enhanced when the tests employ numerous shorter sub-trajectories, specifically length 3. The rationale for this phenomenon is that when the length of the sub-trajectories grows, the quantity of trajectories in the historical data set that may meet the pattern of sub-trajectories of the same length decreases. Consequently, this deficiency of matching data leads to the delivery of empty findings. The accuracy of the prediction approach is influenced by both the size and the number of frequent sub-trajectories. In the subsequent experiment, based on the real data set described in this study, the number of frequent sub-trajectories is established at 20, and the length of these sub-trajectories is set at 3.

### 5.3 Four analyses of the accuracy of store type forecasts.

The order in which a user peruses different types of shops on a website indicates their preferences and intentions for purchasing, as viewed from a profound semantic standpoint. Here, we validate the accuracy of the Markov model in predicting the semantic information shop kinds visited by mobile users in shops. Based on the generated shop category data, the experiments are trained and predicted. Most trajectory sequences in the prediction results of 348 entries had a

**Table 3. Effect of trajectory length on results.**

| Trajectory length | P&1 | P&6 | MRR |
| --- | --- | --- | --- |
| 4 | 0.1345 | 0.3627 | 0.2611 |
| 5 | 0.1355 | 0.3861 | 0.2655 |
| 6 | 0.1376 | 0.3853 | 0.2760 |
| 7 | 0.1387 | 0.3955 | 0.2788 |
| 8 | 0.1414 | 0.4084 | 0.2763 |
| 9 | 0.1562 | 0.4173 | 0.2903 |
| 10 | 0.1442 | 0.4095 | 0.2769 |
| 11 | 0.1438 | 0.4032 | 0.2773 |

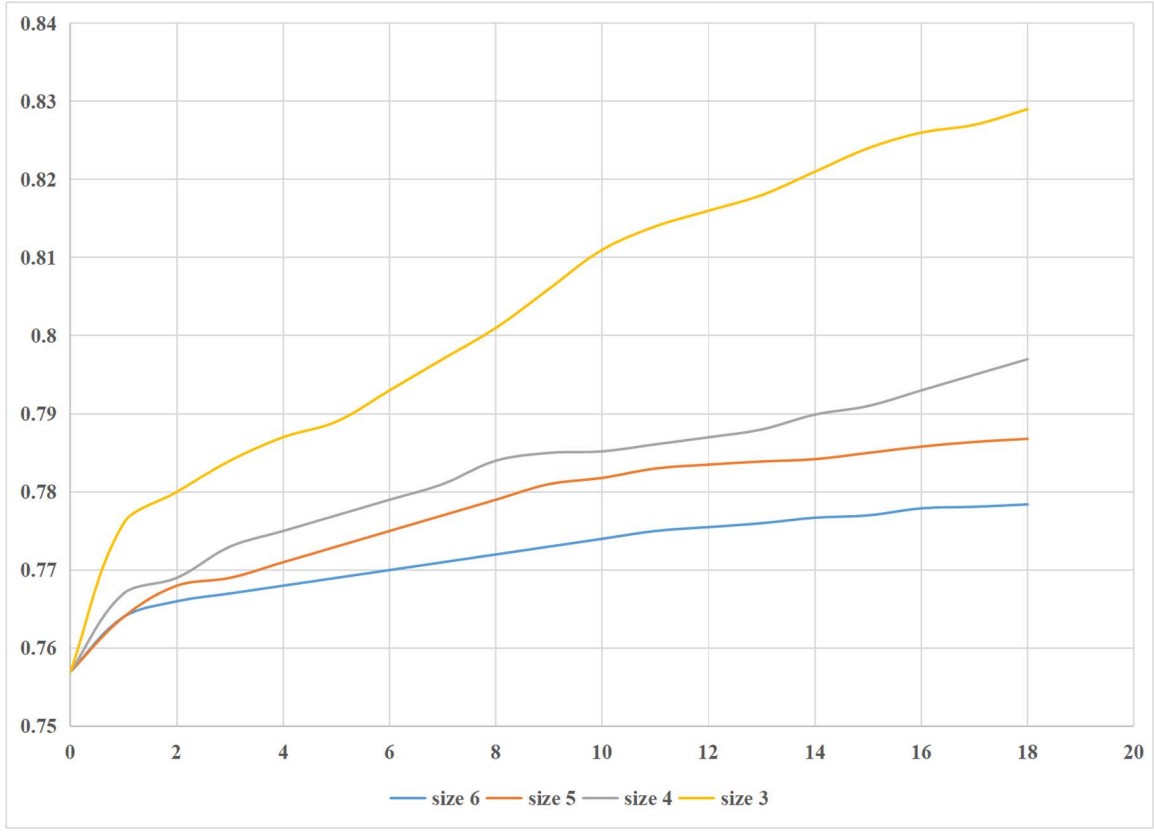

**Fig 4. Effect of the size of the frequentist trajectory on the results.**

concentrated number of trajectory points, ranging from 4 to 11. Approximately 90% of the overall trajectories and trajectories with a length below 4 indicate movement patterns that are too uncomplicated to have a low reference value.

The empirical results indicate that the Markov model achieves an accuracy of shop-type prediction ranging from 13% to 14%, 47% to 49%, and 31% to 32% for the assessment measures p&1, p&6, and MRR, respectively. Thus, it can be inferred that the browsing conduct of consumers in purchasing exhibits a certain level of intentionality. Therefore, it can be assumed that including store type in the prediction will enhance the precision of the model trajectory forecast. The results of training and predicting shop-type sequences of different lengths are shown in Table 4.

Having successfully validated the impact of trajectory length, length and number of frequent sub-trajectories, start time, and dwell time on the prediction accuracy of browsing trajectories, as well as the prediction accuracy of shop types using Markov modeling, this subsection will now proceed to validate the accuracy of the trajectory prediction method taking into account shop types. Fig 5 depicts the outcomes of altering the semantic weighting parameter α on the prediction results.

The provided figure demonstrates that the prediction accuracy of the proposed method increases progressively as the weighting coefficient increases. The prediction accuracy reaches its maximum value when the weighting factor γ is increased. At a weighting factor of 0.8, the prediction accuracy attains its peak value, beyond which a certain decrease level exists. Hence, by considering the specific category of the shop the user intends to visit, it is possible to cut down the pool of potential shops and enhance the significance of each shop. Consequently, this leads to an improvement in the precision of trajectory prediction. Table 5 illustrates the experimental comparison findings of the methods presented in this section with the benchmark methods HMM, LHMM, MMC, HST-LSTM, TPA, and STM.

**Table 4. Impact of shop type prediction results.**

| Trajectory length | P&1 | P&6 | MRR |
|---|---|---|---|
| 4 | 0.1455 | 0.4727 | 0.3114 |
| 5 | 0.1461 | 0.4861 | 0.3134 |
| 6 | 0.1466 | 0.4883 | 0.3278 |
| 7 | 0.1312 | 0.4954 | 0.3131 |
| 8 | 0.1441 | 0.4736 | 0.3071 |
| 9 | 0.1473 | 0.4872 | 0.3122 |
| 10 | 0.1435 | 0.4835 | 0.3161 |
| 11 | 0.1388 | 0.4932 | 0.3143 |

Statistical analysis in Table 5 reveals that HMM and LHMM prediction techniques exhibit the lowest performance. It may be because these two benchmark approaches fail to address the optimal hidden state problem. Instead, they only identify the trajectory points as hidden states, leading to poor forecast accuracy. The Markov model-based prediction algorithms exhibit superior performance, achieving prediction accuracies of 41.6% and 45.6% for the MMC and TPA algorithms, respectively, when evaluated using the p&6 measure. After 260 training rounds, the HST-LSTM model achieves consistent prediction precision. The HST-LSTM model, which utilizes short and long-memory neural networks, achieves prediction accuracies of 14.77%, 37.88%, and 26.04% for trajectories, as evaluated by the metrics p&1, p&6, and MRR, respectively. The trajectory prediction accuracy achieved using LSTM short and long-memory neural networks surpasses the prediction approach based on Hidden Markov Models. However, the marginal advantage is not substantial when compared to the prediction accuracy of the Markov Models-based method.

Furthermore, our proposed trajectory prediction method exhibits much lower prediction accuracy. Compared to the approaches above, the trajectory similarity-based method STM also achieves high prediction accuracy. However, this paper's algorithm significantly enhances the prediction accuracy for the evaluation metrics p&1, p&6, and MRR compared to STM. This is because the prediction method in this paper addresses the issue of sparsity in the transfer probability matrix by proposing frequent sub-trajectories. It solves the problem of predicting the returned result as null and enriches the representation of user behavioral patterns by introducing the concept of dwell time. Consequently, the trajectory prediction effect is improved.

## 6. Conclusion

The research area of user behavior prediction for e-commerce mainly depends on data-driven analysis to develop predictions by analyzing user movement patterns in trajectory sequences. Nevertheless, current approaches have notable deficiencies, such as inadequate analysis of user behavior, complexity in user clustering, constraints in prediction accuracy, and difficulties in computing efficiency and achievement of outcomes.

This paper presents a set of novel strategies to address these difficulties. The first step was a thorough examination of multi-dimensional characteristics. We extensively investigated trajectory data's geographical, temporal, and semantic aspects and compared them with the most advanced prediction techniques already available. Using a density-based clustering approach, we provide a clustering algorithm that utilizes clickstreams and custom events to partition user sessions into distinct clusters. This method aims to enhance the accuracy of identifying user behavior patterns. Furthermore, we present a way to predict trajectories using Hidden Markov Models. This method combines spatial and semantic characteristics and effectively uncovers the best-hidden state of the model by using a similarity clustering algorithm for trajectory points and a transformation algorithm for frequent sub-trajectories.

The next phase of our research endeavors will involve a comprehensive exploration of the following aspects:

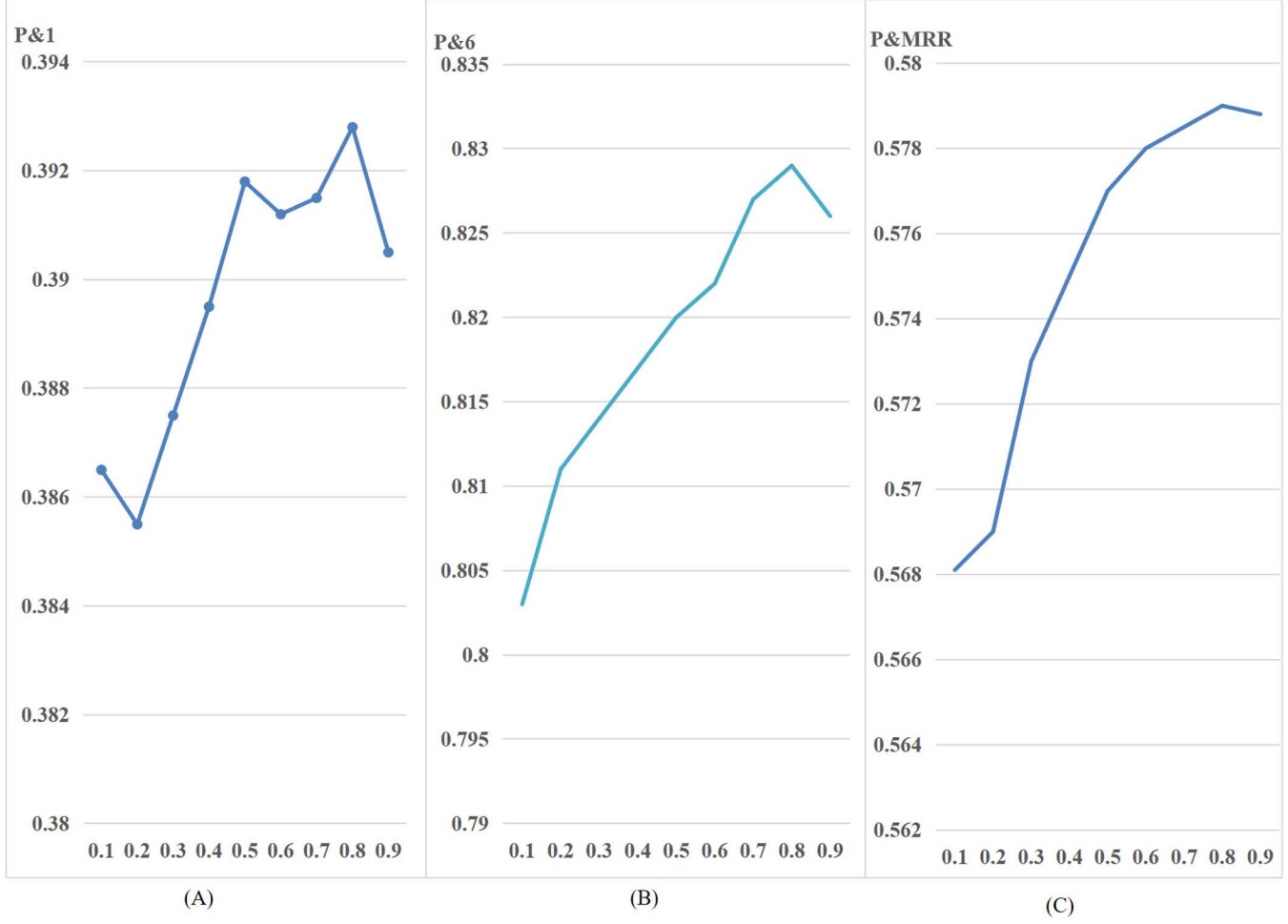

**Fig 5. Effect of weighting coefficients on results.** (A) illustrates how the prediction results change with variations in the parameter α when the trajectory length is $P\&1$.(B) illustrates how the prediction results change with variations in the parameter α when the trajectory length is$P\&6$. (C) illustrates how the prediction results change with variations in the parameter α when the trajectory length is *MRR*.

**Table 5. Impact of the results of the prediction method.**

| Forecasting methodology | P&1 | P&6 | MRR |
|---|---|---|---|
| HMM | 0.0784 | 0.3306 | 0.2133 |
| L HMM | 0.0866 | 0.2970 | 0.2109 |
| MMC | 0.1427 | 0.4166 | 0.2867 |
| HST-LSTM | 0.1477 | 0.3788 | 0.2604 |
| TPA | 0.1598 | 0.4566 | 0.3038 |
| STM | 0.2946 | 0.5988 | 0.4823 |
| Ours | 0.3946 | 0.8398 | 0.5899 |

1. Construction of Long-Term Forecasting Models: Present models mostly concentrate on predicting short-term trajectories without extensive study on long-term or macro-trajectory prediction. Our objective is to create novel models that can effectively capture and forecast patterns in user activity over extended durations.

2. Development of hybrid models: Our study aims to generate hybrid models that integrate the sensitivity of probabilistic statistical models with the high accuracy of deep learning models to improve the precision and robustness of predictions.

3. Enhancement of real-time and accuracy: We will refine our algorithms to enhance real-time predictions and accuracy, particularly in intricate dynamic settings, for real-time application scenarios.

## Author contributions

**Methodology:** Dong-feng Liu.

**Project administration:** Dong-feng Liu.

**Writing – original draft:** Xin Wang.

**Writing – review & editing:** Xin Wang.

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
