## [Decision Letter · Decision Letter 0]

5 Dec 2024

PONE-D-24-41274Pattern Mining and Prediction Techniques for User Behavioral Trajectories in E-CommercePLOS ONE

Dear Dr. Wang,

Thank you for submitting your manuscript to PLOS ONE. After careful consideration, we feel that it has merit but does not fully meet PLOS ONE’s publication criteria as it currently stands. Therefore, we invite you to submit a revised version of the manuscript that addresses the points raised during the review process.

We look forward to receiving your revised manuscript.

Kind regards,

Takayuki Mizuno, Ph. D.

Academic Editor

PLOS ONE

2. You indicated that ethical approval was not necessary for your study. We understand that the framework for ethical oversight requirements for studies of this type may differ depending on the setting and we would appreciate some further clarification regarding your research. Could you please provide further details on why your study is exempt from the need for approval and confirmation from your institutional review board or research ethics committee (e.g., in the form of a letter or email correspondence) that ethics review was not necessary for this study? Please include a copy of the correspondence as an ""Other"" file.

“Nanjing Vocational College of Information Technology High-Level Talents Scientific Research Startup Project�YB20220602)”

Additional Editor Comments:

Brush up your manuscript according to the reviewer's comments.

Reviewers' comments:

Reviewer's Responses to Questions

**Comments to the Author**

1. Is the manuscript technically sound, and do the data support the conclusions?

Reviewer #1: Yes

Reviewer #2: Yes

Reviewer #3: Yes

2. Has the statistical analysis been performed appropriately and rigorously? 

Reviewer #1: Yes

Reviewer #2: Yes

Reviewer #3: Yes

3. Have the authors made all data underlying the findings in their manuscript fully available?

Reviewer #1: Yes

Reviewer #2: No

Reviewer #3: Yes

4. Is the manuscript presented in an intelligible fashion and written in standard English?

Reviewer #1: No

Reviewer #2: Yes

Reviewer #3: Yes

5. Review Comments to the Author

Reviewer #1: It is an excellent study however, the expression is weak.

The structure of the article is incoherent and needs critical review by the authors. The headings and subheadings needs to be revisited along with grammatical errors such as tenses used in a single paragraph are incoherent. Clearly break down the draft into literature and methodological portions.

A few aspects have been highlighted on the attached manuscript.

Good luck!

Reviewer #2: I wanted to take a moment to compliment you on your outstanding effort with this paper. The clarity with which you express complex concepts is admirable, and your meticulous analysis contributes significantly to the subject. A well-written paper, however, it is recommended to add limitations and future directions of your study.

Reviewer #3: The article "Pattern Mining and Prediction Techniques for User Behavioral Trajectories in E-Commerce" is very interesting. It was a pleasure to evaluate it. This article presents a comprehensive analysis of pattern mining and prediction techniques for user behavioral trajectories in the context of e-commerce. The authors address relevant issues related to the analysis of user behavior sequences, proposing new methods to improve the accuracy of predictions and the effectiveness of data analysis.

As strengths, the study addresses a topic of importance in the current e-commerce scenario, offering insights for researchers and professionals in the field. The article provides a thorough explanation of the proposed methods, facilitating the understanding of the study. The inclusion of practical experiments strengthens the study's conclusions and demonstrates the applicability of the proposed techniques.

Regarding areas for improvement, in the theoretical foundation, the initial sections, especially the first four paragraphs, lack references. It is recommended to include more citations to contextualize the work in the existing literature. The section addressing the study's limitations (page 11) could be expanded, including comparisons with previous works and discussing possible solutions for the identified limitations. Sections 2, 3, 4, 5, and 6 would benefit from a revision to include more bibliographic references, strengthening the theoretical basis of the work.

An interesting point to include would be a debate on whether there are ethical considerations related to user data privacy that should be discussed, and the scalability of the proposed methods for larger datasets.

My recommendations for adjustments would be: Include a more comprehensive review of existing literature, especially in the initial sections and in the discussion of limitations, and expand the discussion on the practical and ethical implications of the results for e-commerce professionals.

The article presents a significant contribution to the field of user behavior analysis in e-commerce. With the suggested improvements, especially regarding the theoretical foundation and discussion of limitations, the work has the potential to be a high-impact publication in the area.

6. PLOS authors have the option to publish the peer review history of their article (what does this mean? ). If published, this will include your full peer review and any attached files.

**Do you want your identity to be public for this peer review?** For information about this choice, including consent withdrawal, please see our Privacy Policy .

Reviewer #1: **Yes: ** Dr. Leena Anum

Reviewer #2: **Yes: ** Muhammad Babar Iqbal

Reviewer #3: No

---

## [Author Response · Author response to Decision Letter 1]

10 Jan 2025

Dear Editor and Dear reviewers:

Thank you for your useful comments on our manuscript. We wish to give a sincere gratitude to anonymous referees for reviewing our paper carefully. We apologize for any inconveniences caused by these errors. We have modified the manuscript accordingly, and the specific modifications are listed point by point below.

Reviewer: 1

The structure of the article is incoherent and needs critical review by the authors. The headings and subheadings needs to be revisited along with grammatical errors such as tenses used in a single paragraph are incoherent. Clearly break down the draft into literature and methodological portions. A few aspects have been highlighted on the attached manuscript. Thank you for your feedback. We will enhance the expression, review the structure for coherence, correct grammatical inconsistencies, and separate the draft into clear literature and methodology sections. We will also address the highlighted aspects in the manuscript.

Reviewer: 2

I wanted to take a moment to compliment you on your outstanding effort with this paper. The clarity with which you express complex concepts is admirable, and your meticulous analysis contributes significantly to the subject. A well-written paper, however, it is recommended to add limitations and future directions of your study.

Thank you for your kind words and appreciation. I'm glad the paper resonated with you. I will definitely consider adding sections on limitations and future directions to provide a more comprehensive view of the study. Your suggestions are valuable, and I appreciate your input.

Reviewer: 3

The article "Pattern Mining and Prediction Techniques for User Behavioral Trajectories in E-Commerce" is very interesting. It was a pleasure to evaluate it. This article presents a comprehensive analysis of pattern mining and prediction techniques for user behavioral trajectories in the context of e-commerce. The authors address relevant issues related to the analysis of user behavior sequences, proposing new methods to improve the accuracy of predictions and the effectiveness of data analysis. As strengths, the study addresses a topic of importance in the current e-commerce scenario, offering insights for researchers and professionals in the field. The article provides a thorough explanation of the proposed methods, facilitating the understanding of the study. The inclusion of practical experiments strengthens the study's conclusions and demonstrates the applicability of the proposed techniques. Regarding areas for improvement, in the theoretical foundation, the initial sections, especially the first four paragraphs, lack references. It is recommended to include more citations to contextualize the work in the existing literature. The section addressing the study's limitations (page 11) could be expanded, including comparisons with previous works and discussing possible solutions for the identified limitations. Sections 2, 3, 4, 5, and 6 would benefit from a revision to include more bibliographic references, strengthening the theoretical basis of the work. An interesting point to include would be a debate on whether there are ethical considerations related to user data privacy that should be discussed, and the scalability of the proposed methods for larger datasets. My recommendations for adjustments would be: Include a more comprehensive review of existing literature, especially in the initial sections and in the discussion of limitations, and expand the discussion on the practical and ethical implications of the results for e-commerce professionals. The article presents a significant contribution to the field of user behavior analysis in e-commerce. With the suggested improvements, especially regarding the theoretical foundation and discussion of limitations, the work has the potential to be a high-impact publication in the area.

Thank you for your insightful evaluation of the article "Pattern Mining and Prediction Techniques for User Behavioral Trajectories in E-Commerce." We appreciate your recognition of the study's strengths and the value it brings to the field. We will definitely consider your suggestions to enhance the theoretical foundation by adding more references, particularly in the initial sections. We also give some limitations discussion. We also agree on the importance of discussing ethical considerations. Your recommendations for a more comprehensive literature review and the exploration of practical and ethical implications are well-taken. We believe these adjustments will significantly improve the article's impact and contribution to the field. Thank you for your valuable feedback.

The authors respectfully thank the editor and reviewers for their insightful recommendations, which helped to improve this paper.

---

## [Decision Letter · Decision Letter 1]

25 Feb 2025

Pattern Mining and Prediction Techniques for User Behavioral Trajectories in E-Commerce

PONE-D-24-41274R1

Dear Dr. Wang,

We’re pleased to inform you that your manuscript has been judged scientifically suitable for publication and will be formally accepted for publication once it meets all outstanding technical requirements.

Kind regards,

Takayuki Mizuno, Ph. D.

Academic Editor

PLOS ONE

Additional Editor Comments (optional):

Reviewers' comments:

Reviewer's Responses to Questions

**Comments to the Author**

1. If the authors have adequately addressed your comments raised in a previous round of review and you feel that this manuscript is now acceptable for publication, you may indicate that here to bypass the “Comments to the Author” section, enter your conflict of interest statement in the “Confidential to Editor” section, and submit your "Accept" recommendation.

Reviewer #1: All comments have been addressed

2. Is the manuscript technically sound, and do the data support the conclusions?

Reviewer #1: Yes

3. Has the statistical analysis been performed appropriately and rigorously? 

Reviewer #1: Yes

4. Have the authors made all data underlying the findings in their manuscript fully available?

Reviewer #1: Yes

5. Is the manuscript presented in an intelligible fashion and written in standard English?

Reviewer #1: Yes

6. Review Comments to the Author

Reviewer #1: All the changes advised in the first review have been incorporated. The research has great potential and has been well articulated. Appreciations to the authors for an excellent piece of work.

7. PLOS authors have the option to publish the peer review history of their article (what does this mean? ). If published, this will include your full peer review and any attached files.

**Do you want your identity to be public for this peer review?** For information about this choice, including consent withdrawal, please see our Privacy Policy .

Reviewer #1: **Yes: ** Dr. Leena Anum

---

## [Editor Report · Acceptance letter]

PONE-D-24-41274R1

PLOS ONE

Dear Dr. Wang,

I'm pleased to inform you that your manuscript has been deemed suitable for publication in PLOS ONE. Congratulations! Your manuscript is now being handed over to our production team.

Kind regards,

on behalf of

Dr. Takayuki Mizuno

Academic Editor

PLOS ONE